# Dysfunctional Mechanotransduction through the YAP/TAZ/Hippo Pathway as a Feature of Chronic Disease

**DOI:** 10.3390/cells9010151

**Published:** 2020-01-08

**Authors:** Mathias Cobbaut, Simge Karagil, Lucrezia Bruno, Maria Del Carmen Diaz de la Loza, Francesca E Mackenzie, Michael Stolinski, Ahmed Elbediwy

**Affiliations:** 1Protein Phosphorylation Lab, Francis Crick Institute, London NW1 1AT, UK; mathias.cobbaut@crick.ac.uk; 2Department of Biomolecular Sciences, Kingston University, Kingston-upon-Thames KT1 2EE, UK; k1931747@kingston.ac.uk (S.K.); k1463249@kingston.ac.uk (L.B.); M.Stolinski@kingston.ac.uk (M.S.); 3Department of Chemical and Pharmaceutical Sciences, Kingston University, Kingston-upon-Thames KT1 2EE, UK; F.Mackenzie@kingston.ac.uk; 4Epithelial Biology Lab, Francis Crick Institute, London NW1 1AT, UK; carmen.diaz@crick.ac.uk

**Keywords:** Hippo, YAP, TAZ, polarity, cancer, mechanotransduction, obesity, aPKC, neurodegenerative disease, talin, integrins

## Abstract

In order to ascertain their external environment, cells and tissues have the capability to sense and process a variety of stresses, including stretching and compression forces. These mechanical forces, as experienced by cells and tissues, are then converted into biochemical signals within the cell, leading to a number of cellular mechanisms being activated, including proliferation, differentiation and migration. If the conversion of mechanical cues into biochemical signals is perturbed in any way, then this can be potentially implicated in chronic disease development and processes such as neurological disorders, cancer and obesity. This review will focus on how the interplay between mechanotransduction, cellular structure, metabolism and signalling cascades led by the Hippo-YAP/TAZ axis can lead to a number of chronic diseases and suggest how we can target various pathways in order to design therapeutic targets for these debilitating diseases and conditions.

## 1. Mechanotransduction: Overview and Structural Basis

The ability of cells to detect and react to changes in their microenvironment is critical for the maintenance of both cellular function and tissue homeostasis [1,2]. Recent studies have expanded our understanding of a cell’s ability to sense its physical surroundings and respond to these stimuli to initiate the activation of intracellular signalling pathways that trigger downstream events [1,3]. Indeed, living cells are constantly in contact with mechanical cues and can translate the mechanical forces experienced in their environment into biochemical and behavioral responses. Cells can react to these forces through specific basal proteins and transduce this into a process known as mechanotransduction [2,4,5]. Mechanotransduction is a process initiated by mechanically induced changes in protein conformation which result in cytoskeleton rearrangements, cell division control and differentiation. The mechanical forces experienced by the cell are processed into biochemical signals to ultimately control gene expression, cell shape and cell fate [6]. The cell’s response to mechanical cues within the environment requires the cell’s ability to both transmit and generate forces. This occurs through basal interactions with the cellular extracellular matrix (ECM) and cell-cell interactions with neighbouring cells through specialized transmembrane proteins [1,2]. Mechanotransduction is crucial for many cellular functions such as cell adhesion, motility, migration, proliferation, differentiation and survival. If abnormalities in mechanotransduction occur this can for example lead to aberrant cell behavior and activation of key signalling pathways to drive phenotypical switching into more mesenchymal-like cancer cells from normal cells [4,7].

Cells are physically connected to the ECM through the transmembrane receptors protein family of integrins focal adhesion (FA-related proteins). External mechanical signals are sensed at FAs and translated into biochemical information through integrin-related signalling pathways for maintaining proper cellular functions in cell shape, migration and survival [8]. Integrin activation is required for the normal functionality of cell adhesion, migration and extracellular matrix assembly. Integrins are a family of heterodimeric proteins comprising alpha and beta subunits that are activated through modulation of the cytoplasmic tail of integrin subunits and this activation is regulated by various biochemical signalling pathways. Besides their requirement for normal cellular mechanosensing, tumor cell migration, invasion, growth and metastasis are also regulated by abbrerant integrin-mediated mechanical signals [9,10].

One of the most extensively studied set of oncogenes of recent times and a potent mechanosensor in mammals are Yes-associated protein (YAP) and its homologous protein Transcriptional Co-Activator With PDZ-Binding Motif (TAZ). YAP/TAZ are sensors of structural and mechanical features of the cell microenvironment that are regulated by soluble extracellular factors, cell-to-cell adhesion and mechanotransduction [11,12] and play a central role in delivering mechanical cues from surrounding cells to the transcriptional machinery of the nucleus [13,14]. YAP and TAZ function as transcriptional co-activators shuttling between the cytoplasm and nucleus to induce the expression of cell-proliferative and anti-apoptotic genes by interacting with transcription factors, especially the TEA domain family members (TEAD) [12]. Although YAP is cytoplasmic at high cell density, the loss of cell contact with neighbouring cells or cell spreading results in YAP translocation to the nucleus. The expansion of cell contacts spreading over their basal substrate, the strength of the substrate and resulting mechanical tension are the key determinants of subcellular YAP localization in response to cell density [11,15,16]. Mechanotransduction affects YAP both independently of the canonical Hippo signalling pathway and with the involvement of key Hippo signalling pathway kinases MST and LATS. Signalling cascades downstream of integrins are known to regulate YAP and TAZ activation [17,18,19]. Most of the main mechanosensitive phenotypes induced through YAP/TAZ lean on integrin-mediated adhesion [11]. YAP and TAZ can also be directly regulated by the extracellular matrix (ECM) stiffness and YAP itself is involved in FA assembly [15,20,21] (Figure 1).

Mechanosensing the alterations of ECM stiffness can also be a causal input driving aberrant cell behaviours. The stiffness of ECM is a universal mechanical cue which control’s the preference of a cell between its choice of proliferation and death [16]. When cells perceive stiffness in the ECM, they change their spread and migration potential accordingly. YAP/TAZ activity is also regulated in this scenario. As mentioned, YAP and TAZ are activated by a stiff ECM and stretched cell shape in contrast to inhibition by a soft ECM environment and round cell shape [11,14,16,22,23]. Cells cultured on highly stiff ECM display activated YAP/TAZ with high nuclear localization and subsequent transcriptional activity, whereas cells arrest when YAP/TAZ is inhibited and relocalised to the cytoplasm on a soft ECM. In stiffer ECM, cells continue to proliferate and execute invasive phenotypes by increasing the expression and activity of adhesion receptors and thereby mechanotransduction pathways [11,14,15,16]. Another key regulator of mechanotransduction in relation to its environment is cellular polarity, which we will discuss in the next section.

## 2. Mechanotranduction and Cell Polarity

Cell polarization is an essential function of several cell types including epithelial cells and migratory cells. This part of the review will focus on the interplay between cell polarity and mechanical cues. Cell polarity can occur along several axes, including an anterio-posterior (A–P) axis (e.g., in migratory cells and neurons), and apicobasal and planar axes (e.g., in epithelia). In epithelia and other cell types, apico-basal polarity is the consequence of localization and activity of mutually antagonizing complexes, including the Par, Crumbs and Scribble complexes (for reviews see [24,25]). The Par proteins are segregated along the apico-basal axis in epithelia with an apical Par complex consisting of Par6/Par3/aPKC and a basolateral complex consisting of Par1/4/5. At the apical membrane aPKC activity functions to activate apical polarity substrates and restrict activity of basolateral proteins and conversely, Par-1 mediates kinase activity at the basolateral membrane where aPKC is excluded [26]. In the *C. elegans* zygote, a similar distribution of Par proteins is observed along an A/P axis. Here polarity is a result of mechanical cues exerted by cytoskeletal components. Indeed, the differential diffusion of Par proteins is a consequence of actomyosin contractions orchestrated by the oocyte maturation; coinciding with a cue originating from the centrosome, which causes a cortical flow [27,28]. This flow redistributes Par3 to the anterior side via a process called advection. The activity of the redistributed Par proteins in this manner allows for chemical amplification of A/P identity and polarization. Mechanical cues are sufficient for other cells to polarize along an axis as well. For example, in an attempt to find the minimal requirement for polarization of fibroblasts it was recently shown that a single point of adhesion with an above-threshold force to a fibronectin coated bead is enough to cause redistribution of the actomyosin structure and repositioning of the centrosome, supporting polarized behaviour [29,30]. In migratory cells mechanical force also induces polarization by formation of a single leading edge and suppressing of other protrusions by means of sensing membrane tension. This has been elegantly shown in neutrophils, where a leading edge protrusion characterized by increased membrane tension exerts long range inhibition on the rest of the cell, as measured by reduced Rac activity and SCAR/WAVE complex formation. This long-range inhibition is purely the result of a mechanical stimulus since application of membrane tension by applying a sucking force with a micropipette at one end of the cell causes the described effects [31].

Polarity proteins have also been involved in mechanosensing and altering the mechanical properties of cells. For example, aPKC activity has been linked to remodelling of the keratin intermediate filament (KIF) network in lung epithelial cells upon shear stress by phosphorylation of keratin at the phosphosite Ser-33, promoting its interaction with 14-3-3 [32]. Furthermore, aberrant aPKC signalling is oncogenic and elevated expression levels of aPKC have been observed in several cancers [33,34,35,36]. Besides loss of polarity, overexpression of aPKC in cancer has also been implicated in altering the mechanical properties of cells. Indeed, spheroids consisting of MCF10A breast cancer cells expressing aPKC have been observed to have increased surface tension [37]. Furthermore junctions between aPKC expressing and non-aPKC expressing cells bear an increased tension that allow them to actively extrude into the lamina [37].

Localization of polarity proteins also induces changes in local cytoskeletal composition. For example, Rho-GTPases such as Cdc42 and Rac stimulate actin nucleators and suppress destabilizing factors. Additionally, dynein pulling motors are regulated by the PAR polarity proteins in A/P polarized cells, where inhibition occurs at the anterior pole, resulting in pulling of the mitotic spindle towards the posterior pole [38,39]. Specifically aPKC has been implicated in phosphorylation of LIN5 inhibiting microtubule pulling force [40].

Mechanical cues and polarity have also been interlinked in higher order morphogenesis. For example in the *C. elegans* embryo, planar polarity of the apical Par module in the epidermis is established through muscle contractions that are relayed to the epidermal tight junctions via integrins and ECM components and hemidesmosomes [41]. Coupling of polarity to altering mechanistic properties has also been observed in fly follicles, where egg chamber rotation and elongation are an important part of maturation. Cetara and co-authors show in a recent paper that the elongation of the egg chamber is dependent on the planar alignment of actin fibers at the basolateral membrane [42]. They further show that loss of egg rotation disturbs this planar alignment, likely due to the fact that polarity information cannot be transmitted to the expanding epithelium surrounding the egg, but the exact mechanism has not been elucidated.

Several polarity proteins also induce signalling to the Hippo pathway and YAP/TAZ itself. It has been shown that the apical transmembrane protein Crumbs can associate with FERM domain-containing protein 6 (FRMD6), Kibra and Merlin composing a scaffold that activates the Hippo pathway [43,44,45]. Furthermore YAP is also supressed by the tight junction binding protein AMOTS [46,47,48] and by binding to α-catenin at adherens junctions [49,50].

In contrast, aPKC and the Par complex have been shown to drive nuclear YAP signalling. In epithelial cells overexpression of activated aPKC causes loss of contact inhibition and overgrowth by uncoupling MST1/2 from LATS1/2, causing nuclear accumulation of YAP. Additionally in MDCK cells specifically, downregulation of AMOTS expression also caused a loss of YAP localization to junctions and a spindle like phenotype [51]. Inactivation of the Hippo pathway has also been associated with elevated expression of Par-3 in metastatic prostate cancer by driving a KIBRA/Par-3/aPKC complex formation in lieu of the activating KIBRA/Merlin/FRMD6 complex [52]. Suppression of Hippo signalling by polarity is also seen in the developing embryo, where an inner cell mass is non-polarized and has a different fate then the surrounding trophectoderm which is polarized. This is achieved by polarity signals that inactivate Hippo signalling and cause translocation of YAP to the nucleus, activating a TEAD-dependent transcriptional program (for a recent review see [53]). The Par complex seems to be important for at the 32-cell stage of embryonic development, whereas at the 16-cell stage Par-independent mechanisms are also involved [54].

There seems to be a reciprocal interplay between polarity and Hippo signalling since a recent genetic screen looking for proteins in *C. elegans* that are involved in maintenance of cell polarity in the growing intestine implicated several components of the Hippo pathway including YAP and TEAD [55].

It thus seems that the regulation of Hippo signalling by polarity is dependent on the order of events during polarization, the different polarity complexes formed and thus the level and molecular tuning of the respective proteins, a regulation which seems to include a feedback control from the Hippo pathway itself. How these polarity cues are linked to mechanical signals that act on the Hippo pathway needs to be elucidated further, but because of the close interplay between the two, some cross-regulation is inevitable.

In conclusion it is emerging that mechanical stimuli and polarity are closely interlinked, both at the level of individual cells as well as at the level of the developing tissue. While some systems like the *C. elegans* zygote have been extensively studied, many of the mechanisms regarding coupling of mechanical to chemical stimuli have yet to be resolved. Loss of polarisation can lead to cellular transformation and epithelial to mesenchymal transition, a tumour-initiating process. The next section will explore the link between mechanotransduction and disease.

## 3. Mechanotransduction Signalling in Disease

Mechanotransduction is an important cellular hallmark in the formation of chronic disease. Loss of polarity or mechanosensing can lead to a range of diseases including cancer, obesity and neurodegenerative diseases and various pathways can be involved in this process. Figure 2 provides an overview of the types of signalling pathways involved in these diseases and we shall discuss this in more detail in the coming section.

### 3.1. Cancer

Solid tumors exhibit increased ECM stiffness and crosslinking that promote an invasive phenotype in cancer cells. Increased stiffness of the ECM can also lead to a disruption of the epithelial polarity resulting in migration and metastasis [4]. In addition, stiff ECM activates the β1-integrin-FAK-Src-PI3K-PDK1 pathway to facilitate YAP nuclear translocation and transcriptional activation as previously discussed [11,56,57]. Concomitantly, YAP/TAZ is overexpressed and highly accumulated in the nuclei of cancer cells, initiating the transcription of YAP/TAZ target genes to drive proliferation, invasion and metastasis [13]. Thus, YAP/TAZ plays a crucial role in driving tumorigenesis as dysregulation of the Hippo pathway and sustained activation of YAP drives uncontrollable growth [58]. There is scope for further investigations in the future of the role of novel ECM components and how they relate to Hippo pathway signalling. We have already previously identified a family of ECM cross-linkers (Enigma family members) which link the ECM to the cytoskeleton and result in the regulation of YAP [16]. It has also been shown that the mechanical regulation of the Hippo pathway is conserved in a wide range of species [59,60,61]. Mechanotransduction could play a potentially important role in the development of targeted therapies targeting the cancer cell’s pro-proliferative and survival signals stemming from its interaction with the stroma. An example of such a target could be the talin family of proteins.

Extensive biological studies have hinted that the talin family of proteins may play an important role in the transduction of the integrin signal. Talins are a major cytoskeletal protein at focal adhesions that link intracellular networks with the ECM via its coupling with the actin cytoskeleton and membrane integrins [9,10,62,63]. Talins are comprised of a globular head and a long rod domain. The FERM domain within the talin head domain provides a binding site to the cytoplasmic integrin β subunit whereas the C-terminal talin rod containing several vinculin binding sites provides a connection to the actin cytoskeleton and the ECM within the cell suggesting that talin is an important mechanosensitive molecule [2,64,65]. In vertebrates, there are two talin genes: *talin1* and *talin2*. The talin1 protein was found to regulate focal adhesion dynamics, cell migration and invasion; whereas the precise role of the talin2 protein in cancer is not clear. Talin1 is expressed in nearly all tissues whereas talin2 is mainly expressed in the heart, brain, testis and muscles [66]. Talin2 functions distinctly from talin1 in the regulation of cell invasion and is usually localized at large focal adhesions and fibrillary adhesions. In the absence of talin1, talin2 was found to regulate focal adhesion assembly and focal adhesion kinase signalling [67]. Interestingly, the talin2 head domain (TH2) has a higher affinity and a much stronger binding to β-integrin tails than talin1 head domain (TH1) which is important for traction force generation and cell invasion (Figure 1). The strong binding of talin2 with the β-integrin tail produces the aforementioned traction force which goes on to colocalise with invadopodia and encourages invadopodium-mediated matrix degradation. This is a key mechanism for cancer cell invasion [67]. In breast cancer cell lines, talin2 controls production of traction force through its strong binding with β-integrins thus leading to the regulation of various processes including cell migration, invasion, tumor growth and metastasis and potentially oncogenic activation of various proteins [10]. The talin family has also been linked to obesity since talin levels increase with increasing adipose cell size in the growing fat pad. The next section further explores the link between mechanotransduction and adipose regulation.

### 3.2. Obesity

In addition to chemical and hormonal influences, mechanotransduction signals play an important role in adipogenesis. These mechanical cues as well as having an important role in adipocyte development, also play a major role in differentiation to adipogenic lineages [68]. A combination of all of these influences, in addition to distinct precursor populations of adipocytes, together with final tissue location influencing functionality, results in specific depot physiology for the prevailing adipose tissue mass formed [69,70].

The number of adipocyte cells in human adults appear to be relatively fixed, even after weight loss, with large variations in cell size both within and between individuals. A stable number of cells and an approximate 10% turnover annually, which is not influenced by energy balance, indicates a tight regulatory control of the accumulated tissue mass formed [71,72]. This is of importance in the context of obesity and the resulting pathology caused by dysregulation of whole body metabolism and insulin resistance. The physiological response to excess calorie intake is adipocyte lipid loading, causing an increased cell volume. This propagates the metabolic complications of obesity. Indeed, enhanced accumulation of lipid may lead to adipocyte hypertrophy, characteristic of adipose tissue dysfunction [73].

Adipocytes store lipids in the form of triacylglycerides in a single, large unilocular lipid filled droplet. The consequence of this is an increase in the stiffness of the adipocyte [74]. This stiffness adds to mechanical stress such as different loading or weight bearing strains which also generate forces which adipocytes are exposed to. The resultant mechanical forces can be transduced to affect adipogenesis [75]. These forces are transduced by various protein complexes such as: focal adhesions interconnecting with the actin cytoskeleton [76] or ion channels that are sensitive to mechanical stress [77]. In line with this, a recent study has identified a volume sensitive ion channel on the adipocyte surface which can regulate insulin signalling [78]. This protein, SWELL1, is increased in human hypertrophic adipocytes of obese humans and is required for glucose uptake via PI3K-AKT2-GLUT4 signalling activity. This provides evidence of a link between mechanoreception and insulin signalling which is activated under obesogenic conditions. Following transduction, static or dynamic forces which alter adipocyte cell shape or the adipocyte microenvironment have been shown to activate various intracellular pathways including RhoA/ROCK, Akt/Erk and the Hippo pathway which influence gene expression and cell fate [79].

Chronic mechanical force such as that exerted through extracellular matrix stiffness, as occurs through the development of adipose tissue fibrosis, has been associated with metabolic complications in obesity. Evidence for this has been provided through the preparation of 3D adipocyte hydrogel cultures with incorporated decellularized material from the adipose tissue of obese subjects. This induced dysfunctional adipocyte metabolism and increased expression of connective tissue growth factor (CTGF) through the Hippo related YAP/TEAD-dependent pathway [80].

A cells metabolic status plays an important role in regulating key members of the Hippo pathway, YAP and TAZ. These factors have both been shown to regulate and be regulated by multiple metabolic pathways and energy-sensing pathways such as via AMP-activated protein kinase (AMPK) [81,82,83,84]. The master regulator of metabolism and proliferation, c-Myc, which itself is influenced by growth factor and nutrient availability, has been shown to be influenced by components of the Hippo pathway [85]. It has been shown that adipocyte proliferation may be activated by the SIRT1/c-Myc pathway. Reduced levels of SIRT1 resulted in hyperacetylated c-Myc which promoted the formation of a metabolically dysfunctional 3T3-L1 preadipocyte phenotype [86]. Within adipocytes, the activity of one of the key transcriptional factors for adipogenesis, PPARy is suppressed by TAZ. In this regard, adipose-specific TAZ knockout mice show improvements in insulin sensitivity and glucose metabolism, establishing a potential role for TAZ in promoting insulin resistance [87]. Liraglutide, a GLP-1 analogue and antidiabetic agent also used to treat obesity, has shown to activate the Hippo pathway by increasing YAP related pathway components (MST1, LATS1 and p-YAP). This results in reduced cell proliferation but increased cell differentiation of 3T3-L1 preadipocytes [88]. These studies suggest crucial connections with the regulation of substrate metabolism and the Hippo pathway. It is therefore key to establish the mechanisms for cross-talk between mechanotransduction pathways such as the Hippo pathway and substrate metabolism, tissue expansion, the development of adipocyte hypertrophy and insulin resistance.

Obesity is a risk factor for cancer development, and a recent research focus has provided evidence that this link may be partially due to alterations in the Hippo pathway. Of interest is that adipocytes form a major component of the tumour microenvironment in for example breast, colon and gastric cancers and can therefore directly contribute to cancer cell development [89,90]. Obesity has been shown to accelerate cancer development following the subcutaneous injection of murine forestomach carcinoma cells. This was shown to act through the influence of the adipokine visfatin whose concentration in serum was correlated with SIRT1 (a multifunctional protein that regulates metabolic activity). The upregulated SIRT1 promoted YAP2/TEAD4 activation and the development of cancer in a diet-induced murine obesity model [91]. Obesity and insulin resistance are also important risk factors for the development of endometrial cancer where knockdown of YAP/TAZ has been shown to inhibit insulin signalling via phosphorylation of IRS1/2, a mediator of insulin and IGF-1 signalling in endometrial cell lines [92]. The Hippo pathway may therefore represent a novel target for cancer prevention where adipocytes are closely associated with cancer such as breast cancer [93,94] or where insulin resistance is associated with cancer progression [95].

One strategy for treating obesity has been through inducing a switch from white to brown like (beige) adipocytes, which can arise postnatally in white adipose tissue depots. Beige adipocytes, which dissipate energy as heat as opposed to storing fat, can be induced by various cues including temperature, exercise, adrenergic stimulation, nutritional and hormonal signals [96]. These adipocytes are associated with increased energy expenditure and improved insulin sensitivity, similar to brown adipocytes. Understanding how beige selective pathways are induced can provide new insights that could allow for the targeted selection of this metabolically favorable phenotype of adipocyte. Although developmentally of different origin (muscle-like), YAP/TAZ signalling has also been shown to regulate insulin signalling in muscle where TAZ has shown to stimulate insulin sensitivity by upregulating IRS1 which mediates GLUT4 uptake of glucose in muscle cells [97]. Brown adipocytes (BAT) are functionally similar to beige adipocytes. Both cell types show multilocular deposition of lipid and increased oxidative metabolism through upregulation of UCP1, which induces thermogenesis. It is therefore of interest that BAT development is dependent on YAP/TAZ signalling which upregulates UCP1 expression and that this process is mediated through a cytoskeletal mechanotransduction response [98].

Understanding the role of adipocytes in the context of obesity related disease processes, including cancer, and uncovering tissue-specific molecules related to the functioning of the Hippo pathway could provide biomarkers or targets for therapy. However, the association of the Hippo pathway with multiple functions associated with growth and development in tissues also provides a challenge with respect to potential side effects.

### 3.3. Neurodegenerative Disease

The sensory processes of hearing, touch, and pain sensation depend on transducing certain types of mechanical force into sensory information. Sensory disorders involving dysfunction of these processes have been identified (such as deafness) [99]. However, additionally, sensing and transduction of other local tissue mechanical forces are critical for normal neuronal development, homeostasis, and function. In the developing embryonic nervous system, neuronal progenitors migrate, and neuronal axons extend, branch, and find their synaptic targets through various embryonic tissue types [100]. In normal adult function, neurons are also under constantly changing mechanical forces due to the movement or pressure of surrounding muscles or tissues, especially during tissue damage. Stretching of neurons, such as during regenerative axon growth, is known to promote neurite outgrowth and to be a potent regulator of axon growth.

Much is now known of the general molecular and biochemical pathways governing neuronal development, function and disease. However, the specific roles of mechanical forces in neuronal homeostasis and function have only recently started to be determined. For example, retinal ganglion axons prefer to grow through ‘softer’ tissue and avoid ‘stiffer’ tissue in *Xenopus* brains [101]. A wide variety of proteins with links to mechanotransduction have been identified as important to neuronal development and function/homeostasis, including mechanosensitive Piezo ion channels (e.g., Piezo-1, Piezo-2), transient potential receptor (TRP) channels (e.g., TRPV2, TRPV4), transmembrane or extracellular matrix (ECM) proteins (e.g., integrins, β-spectrin), and intracellular signalling proteins (e.g., YAP/TAZ).

Despite this, even less is known of the importance or role of mechanotransduction in the development or progression of neurodegenerative diseases (NDD). Changes in the ECM and/or tissue stiffness have been reported in traumatic brain injury and brain cancer [102] and in central nervous system glial scars in rats [103] but also in Alzheimer’s disease [104,105], Parkinson’s disease [106], and multiple sclerosis [107], suggesting that stiffness changes may be relevant to NDD. Several recent studies have provided intriguing evidence that mechanotransduction proteins may also have fundamental direct roles in neurodegenerative disease.

Piezo-1 is a mechanically gated calcium channel mainly expressed on the cell membrane of myelinated neurons and, as it responds to mechanical stimuli, it plays an important role in mechanotransduction [108]. On activation by traction forces, Piezo-1 allows the entrance of calcium ions (Ca^2+^) into the cell. Activation of Piezo-1 in neuronal stem cells induces their differentiation into neurons, rather than astrocytes [109]. Furthermore, Piezo-1 activation results in the nuclear targeting of YAP and TAZ, an event implicated in the myelin formation in the central nervous system (CNS) [109]. Disorders involving progressive demyelination (loss of myelin), called demyelinating disorders can lead to axonal injury, neurodegeneration, and significant disability and death [110,111]. Demyelinating disorders are often characterised by an axonal Ca^2+^ imbalance, resulting in an increased influx of Ca^2+^, thus causing elevated production of reactive oxygen species and excitotoxicity [112].

In a recent study, enhancing Piezo-1 activation with a small-molecule agonist Yoda-1 caused axon demyelination in mouse cerebellar slices, while a pharmacological Piezo-1 inhibitor, GsMTx4, caused increased myelin formation [108]. It was hypothesised that over-activation of Piezo-1 induces increased influx of Ca^2+^ into CNS axons, release of Ca^2+^ from intracellular stores, and activation of calpain, a protease that can cleave talin and modify integrin-mediated cell adhesion [108]. For this reason, blocking Ca^2+^ influx by inhibition of the Piezo-1 Ca^2+^ channel might have therapeutic implications in demyelinating and neurodegenerative disorders. Furthermore, GsMTx4 attenuated axonal damage induced by demyelination and reduced microglial reactivity following activation by LPS (lipopolysaccharide) [108]. Piezo-1 expression in amyloid plaque-reactive astrocytes is also upregulated by peripheral bacterial infection in rats [113]. As microglial reactivity and/or other immune mechanisms are thought to facilitate neurodegeneration, and are implicated in Alzheimer’s disease progression, this further suggests the potential beneficial use of Piezo-1-modulating agents in treating neurodegenerative disease.

Further evidence of mechanotransduction changes in neurodegeneration disease through the YAP/Hippo pathway has recently been provided. Alexander disease (AxD) is a rare neurodegenerative disease of primary astrocyte dysfunction which features gain-of-function mutations in and protein aggregations of the astrocytic protein GFAP [114]. A *Drosophila* AxD model overexpressing a human mutant form of GFAP (GFAP^R79H^) showed activation of the YAP/Hippo pathway, with resulting abnormal changes to mechanotransduction proteins including lamin, focal adhesion kinase (FAK) and ECM proteins, leading to behavioural abnormalities and neurodegeneration in this fly model [114]. Furthermore, brains of transgenic AxD mice showed increased tissue stiffness and iPSC-derived astrocytes from one AxD patient displayed increased A-type lamin and nuclear-localised YAP compared to control astrocytes [114]. Interestingly, a neuronal-specific isoform of YAP, YAPdeltaC, protects against neuronal apoptosis in a model of the neurodegenerative disease Huntington’s disease (HD), by blocking interaction of full-length YAP with p73 [115], and was recently shown to modulate neurodegeneration in a mouse model of spinocerebellar ataxia 1 [116]. Additionally, activation of MST1 is increased in human post-mortem HD cortex, and nuclear localisation of YAP is correspondingly decreased in human HD cortex and in neuronal stem cells derived from HD patients (117), leading to altered Hippo pathway gene expression [117]. The YAP/TAZ/Hippo pathway may therefore be a relevant mechanism in mammalian neurodegenerative processes, and perhaps particularly in aggregation-prone neurodegenerative diseases.

Finally, glaucoma is an ocular disease featuring increased intraocular pressure (IOP), which when untreated leads to retinal damage, the irreversible degeneration of retinal ganglion cell (RGC) neurons, and blindness [118]. A number of mechanotransduction mechanisms and proteins have been implicated in the various anatomical and physiological changes in the eye which lead to glaucoma and, as a secondary consequence, RGC degeneration. Interestingly, however, the mechanosensitive ion channels TRPV4 and TREK-1, as well as integrins, have been suggested to be directly involved within RGC in pressure-induced dendritic retraction and RGC loss, in glaucoma [119], through sensing stretch, compression, tension and/or shear of retinal or retinal-associated ocular components. TRPV4 is strongly expressed in RGCs, and induction of TRPV4 causes RGC degeneration and other related ocular pathologies in mice and in vitro [119]. As mice with ablated TRPV4 or TREK genes show reduced and increased sensitivity to mechanical stress, respectively [120,121], these channels may act in opposition to regulate homeostatic responses to mechanical stress in the retina. Interestingly, TRPV4 activation causes YAP/TAZ nuclear translocation during cellular epithelial-mesenchymal transition (EMT) in keratinocytes [122,123]. It will be useful to identify if TRPV4 also effects this process in retinal ganglion cells. Further, aging is a risk factor for glaucoma, as well as other NDD. Many components of the eye undergo tissue stiffening and ECM remodelling in older age [124]. Age-related changes in tissue stiffness may be a relevant avenue for further investigation in NDD development.

In summary, the large number of mechanosensing proteins that have critical roles in neural development and function/maintenance suggests that further roles for these proteins in NDD will likely be discovered.

## 4. Conclusions

As we have conveyed, mechanotransduction plays a huge role in a wide range of cellular and physiological processes of which the dysregulation can contribute to a number of chronic diseases including obesity, diabetes, cancer (including loss of polarity/tissue context) and neurodegenerative disease. Table 1 summarises the key pathways shown in this review, important proteins from these pathways and how these proteins are implicated in a number of diseases.

Many of these signalling pathways involved in the process of mechanotransduction (and thus tissue stiffness) seems to revolve around YAP/TAZ signalling. If YAP/TAZ which are established mechanosensors can be inhibited by therapy then we can potentially affect the levels of chronic disease and tissue stiffening with increasing age.

Future therapeutic options for targeting tissue stiffness will include targeting the Hippo pathway.

As shown in Figure 3, if therapy is initiated during increasing tissue stiffness hypothetically we can reduce the incidence of the chronic disease. The interplay between mechanotransduction and chronic disease is a large field which we can explore with various targeted therapies with the ultimate goal of treating cases of chronic disease.

## Figures and Tables

**Figure 1 cells-09-00151-f001:**
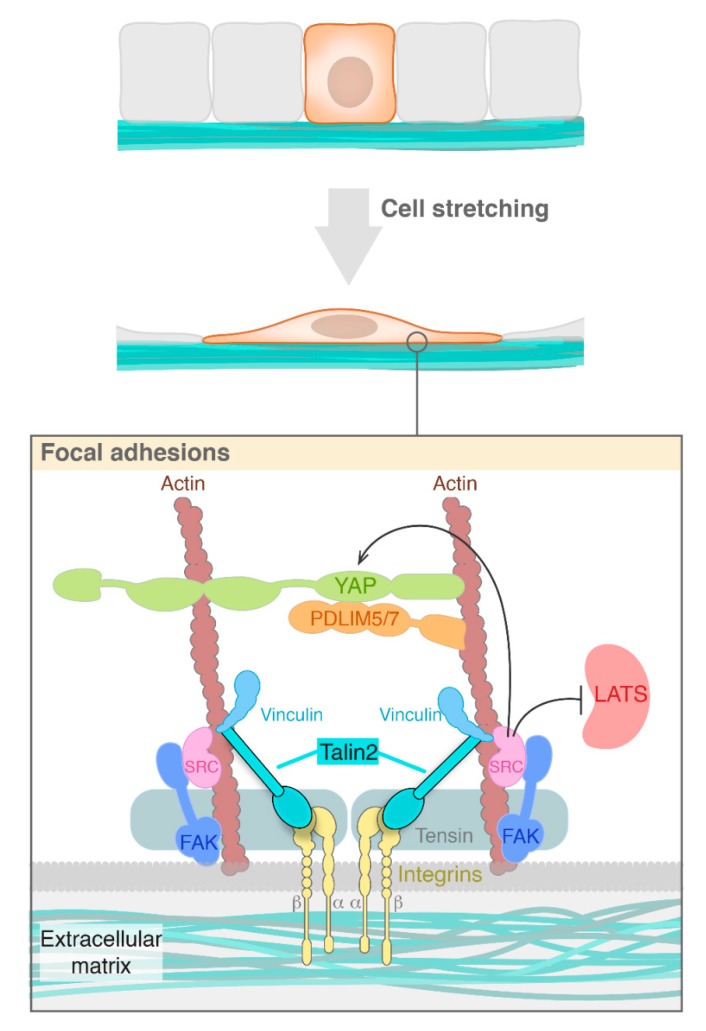
The regulation of YAP at the level of the extracellular matrix involves various proteins involved in stabilizing and activating YAP. Integrins sense the external forces and transmit this signal to talin family members which allow YAP and accessory proteins to act as mechanosensors within the cell.

**Figure 2 cells-09-00151-f002:**
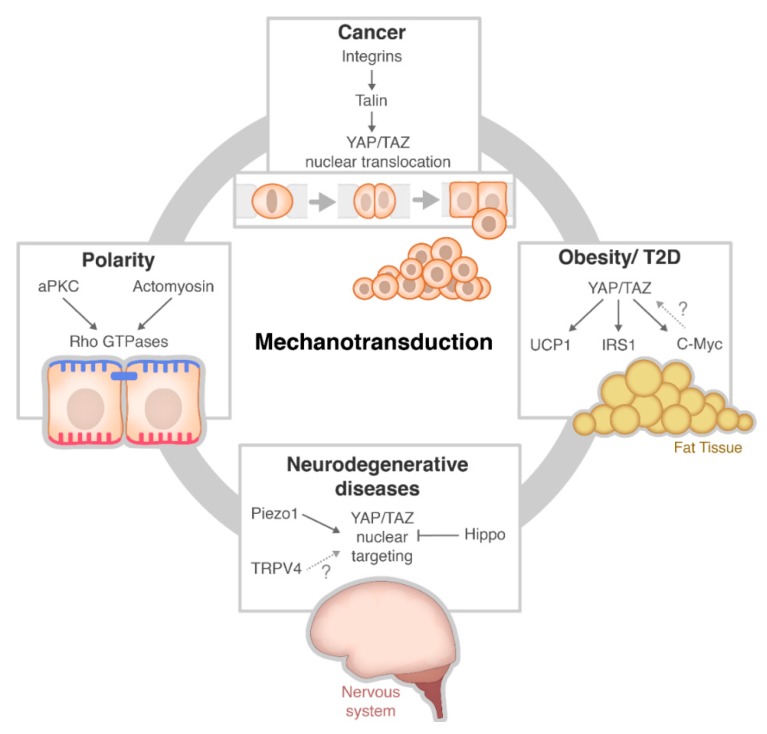
The pathways and interplay with regulate mechanotransduction in the context of chronic disease and polarity. Note that some pathways are not well established and thus represented by a question mark (?) (T2D = Type 2 Diabetes). Please note also → = activates and —| = inhibits.

**Figure 3 cells-09-00151-f003:**
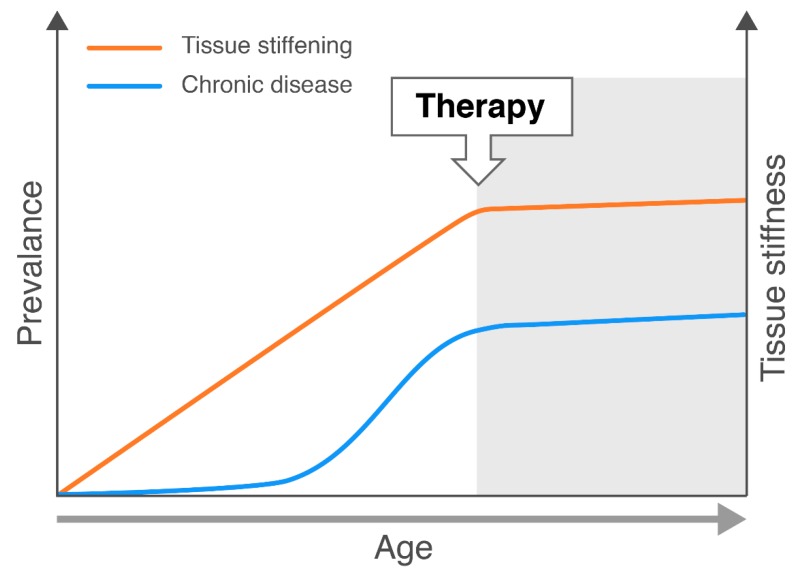
A hypothetical graph linking the effect of tissue stiffness with age and chronic disease. Upon therapy intervention which would target signalling cascades we would assume a correlation between less stiffness and chronic disease.

**Table 1 cells-09-00151-t001:** Proteins discussed in this review involved in mechanotransduction and diseases associated with their dysfunction.

Pathway	Protein	Disease	References
Extracellular matrix (ECM)	Integrins	Cancer	[9,10]
	Talin2	Cancer	[10,66]
Hippo signalling	YAP	Alexander disease	[114]
		Huntington’s disease	[115]
		Spinocerebellar ataxia 1	[116]
		Demyelinating disorders	[109]
		Cancer	[4]
	TAZ	Cancer	[13]
		Demyelinating disorders	[109]
		Type 2 diabetes	[87]
Ion channels	Piezo-1	Demyelinating disorders	[108]
		Alzheimer’s disease	[113]
	TRPV4	Glaucoma	[119]
Polarity	aPKC	Cancer (contributing)	[37]
Metabolic pathways	UCP1	Obesity	[98]
	IRS1	Cancer	[92]

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
