# Peer review of "Dysfunctional Mechanotransduction through the YAP/TAZ/Hippo Pathway as a Feature of Chronic Disease"

_cells, 2020, doi:10.3390/cells9010151_

Round 1

Reviewer 1 Report

This review article describes mechanotransduction and how defects in related biological pathways may lead to different diseases, ranging from obesity to cancer.

The authors have cited a large number of studies, which render this article a good source for researchers to find original studies in this field.

For readers that are not familiar with mechanotransduction, they would appreciate a more detailed summary of all cellular pathways related to this field. In this current manuscript, when the authors describe how genes and biological components are related to diseases, it can be difficult to follow how these genes are involved in the underlying mechanism of the diseases.

There is only one typo I noticed:

Line 76, "Figure shows the way in......" should be "Figure 1 shows the way......".

Author Response

This review article describes mechanotransduction and how defects in related biological pathways may lead to different diseases, ranging from obesity to cancer.

The authors have cited a large number of studies, which render this article a good source for researchers to find original studies in this field.

For readers that are not familiar with mechanotransduction, they would appreciate a more detailed summary of all cellular pathways related to this field. In this current manuscript, when the authors describe how genes and biological components are related to diseases, it can be difficult to follow how these genes are involved in the underlying mechanism of the diseases.

We thank you for this insightful comment and the reviewer is entirely corrected as to how the genes relate to specific diseases, we have now included a table at the end of the review listing the type of gene, pathway, which disease it is linked to and the reference which refers to this.

There is only one typo I noticed:

Line 76, "Figure shows the way in......" should be "Figure 1 shows the way......".

This has been corrected and the manuscript proofread by all the authors to remove any other potential typos.

Reviewer 2 Report

The manuscript is a through review of mechanotransduction.  It would however benefit from an introduction that lays out the flow of the review, so it is easy for the reader to follow. A formal reorganization of the sections with at least 2 overarching sections: 1) background on mechantransduction and stimulated pathways and 2) Mechanotranduction in disease will make it flow better.  Figure 2 and 3 would work nicely in an introduction of the disease model section.  Your conclusion could then discuss how mechantranduction could be a new target for treatment development.  These changes would make your review more impactful.

Author Response

Reviewer 2:

The manuscript is a thorough review of mechanotransduction.  It would however benefit from an introduction that lays out the flow of the review, so it is easy for the reader to follow. A formal reorganization of the sections with at least 2 overarching sections: 1) background on mechantransduction and stimulated pathways and 2) Mechanotranduction in disease will make it flow better.  Figure 2 and 3 would work nicely in an introduction of the disease model section.  Your conclusion could then discuss how mechantranduction could be a new target for treatment development.  These changes would make your review more impactful.

We thank the reviewer for their comments which will make the review stronger. We have now separated the review into two overarching sections:

Mechanotransduction: overview and structural basis and Mechanotransduction signalling in disease. The conclusion has also been expanded to mention how mechanotransduction could be a target for disease.

Reviewer 3 Report

In current manuscript, Cobbaut et al. reviewed the signaling pathways in mechanotransduction and their roles in cell fate decision as well as the process of pathogenesis. The topic is of great interest and the manuscript is well organized. The authors adequately summarized and discussed some of major players and the regulatory mechanisms under different contexts. I do have some minor points that need to be addressed:

1) It would be helpful to have a table to categorize (integrins, ion channels, growth factor receptors, eta.) the major mechanotranscudtion pathways and list their essential components, as well as what diseases they have been reported to be associated with.

2) Figure 2 will be more informative if the pathways can be organized in a way that shows the relationship between them. For example, under obesity, YAP/TAZ regulates the expression/activity of UCP1 and IRS1. While for Cancer, YAP/TAZ is downstream of integrins. In NDD, YAP and TRPV4 most likely are independent of each other.

3) If the review aims to summarize all the major mechanotransduction signals rather than focusing the YAP/TAZ (which is fine, but need to rephrase the title and abstract, etc.), studies on other important pathways, such as WNT and TGF, should be discussed.

Author Response

In current manuscript, Cobbaut et al. reviewed the signaling pathways in mechanotransduction and their roles in cell fate decision as well as the process of pathogenesis. The topic is of great interest and the manuscript is well organized. The authors adequately summarized and discussed some of major players and the regulatory mechanisms under different contexts. I do have some minor points that need to be addressed:

1) It would be helpful to have a table to categorize (integrins, ion channels, growth factor receptors, eta.) the major mechanotranscudtion pathways and list their essential components, as well as what diseases they have been reported to be associated with.

We thank the reviewer for their kind comments, we have now added a table at the end of the review to summarise the type of gene, pathway, and disease associated with the gene in question and reference to this.

2) Figure 2 will be more informative if the pathways can be organized in a way that shows the relationship between them. For example, under obesity, YAP/TAZ regulates the expression/activity of UCP1 and IRS1. While for Cancer, YAP/TAZ is downstream of integrins. In NDD, YAP and TRPV4 most likely are independent of each other.

Figure 2 has now been amended to reflect how the proteins act upon each other.

3) If the review aims to summarize all the major mechanotransduction signals rather than focusing the YAP/TAZ (which is fine, but need to rephrase the title and abstract, etc.), studies on other important pathways, such as WNT and TGF, should be discussed.

We have amended the title to reflect the involvement of YAP/TAZ to now read ‘Dysfunctional mechanotransduction through the YAP/TAZ/Hippo pathway as a feature of chronic disease, we have also amended the abstract and added a paragraph to the polarity section to reflect this.

Round 2

Reviewer 2 Report

Manuscript is a through review of YAP TAZ based mechanotransduction and is suitable for publication in its current form.